# JMJD3 regulates the M2-like macrophage polarization and promotes the growth of breast cancer cells via STAT6/IRF4 axis

Juan Lyu[1,2], Enqin Wang[3], Shanmei Lyu[1,2], Ying Qian[1,2], Fen Ye[1,2], Xiuping Xu[1,2], Qing Wang[1,2], Tao Lu[1,2], Liangfeng Hu[1,2], Hongkun Xu[1,2], Lihong Zhang[1,2*]

**1** Department of Clinical Laboratory, Shaoxing People's Hospital, Shaoxing, Zhejiang, P.R. China, **2** School of Medicine, Shaoxing University, Shaoxing, Zhejiang, P.R. China, **3** Department of pathology, Wuchuan County People's Hospital, Zunyi, Guizhou, P.R. China

* whb0575@163.com

## Abstract

M2-like macrophages play a critical role in breast cancer progression. Although JMJD3 is reported to play a significant role in M2-like macrophage polarization, its precise mechanism remains unclear. By using PMA, IL-4, and IL-13, we successfully induced THP-1 cells into M2-like macrophages, which subsequently promoted breast cancer cell proliferation and inhibited apoptosis, accompanied by increased JMJD3 expression. We demonstrated that JMJD3 enhances M2-like macrophage polarization: knockdown of JMJD3 decreased the M2-like macrophage gene expression, while overexpression of JMJD3 produced the opposite effects. Furthermore, JMJD3 promoted M2-like macrophage polarization through the STAT6/IRF4 axis. Knockdown of JMJD3 abrogated IL-4/IL-13 induced IRF4 expression, while overexpression of JMJD3 upregulated IRF4 expression. Inhibition of STAT6 downregulated the expression of JMJD3, IRF4, and M2-like macrophage marker genes. Additionally, inhibiting JMJD3 and STAT6 in macrophages increased cell apoptosis and decreased cell viability in breast cancer cells, while JMJD3 overexpression exhibited pro-tumor activity. In conclusion, our findings highlight the role of JMJD3 in regulating M2-like macrophage polarization and its impact on breast cancer development through the STAT6/IRF4 axis.

## Introduction

Breast cancer has become the most prevalent cancer and the leading cause of cancer-related deaths among women worldwide [1,2]. Despite advances in screening and treatment, drug resistance and metastasis in breast cancer continue to result in significant patient mortality [3].

Tumor development is closely associated with the tumor microenvironment, which is predominantly composed of macrophages. These macrophages can be polarized

**Data availability statement:** All relevant data are within the manuscript and its Supporting Information files.

**Funding:** The study was supported by the Zhejiang Provincial Natural Science Foundation of China under Grant No. LQ24H200001(to J.L.), Medical Science and Technology Project of Zhejiang Provincial Health Commission under Grant Nos. 2023KY353 (to J.L.) and 2024KY462 (to Y.Q), and by the Shaoxing City Health Science and Technology Plan under Grant No. 2022SY020(to J.L.). The funders had no role in study design, data collection and analysis, decision to publish, or preparation of the manuscript.

**Competing interests:** The authors have declared that no competing interests exist.

into two subtypes: M1-like and M2-like [4]. M1-like macrophages inhibit tumor growth, while M2-like macrophages promote vascularization, immune suppression, and tumor progression. M2-like macrophages are abundant in tumor tissues [5] and are associated with poor prognosis and tumor progression [6–10]. In breast cancer, tumor-associated macrophages (TAMs) comprise 50–80% of the stromal cells. M2-like TAMs promote breast cancer cell proliferation, metastasis, and immunosuppression, angiogenesis, and chemotherapy resistance [6,11–13]. Modulating macrophage polarization may present a novel strategy for treating breast cancer.

JMJD3 (Jumonji domain-containing protein 3) is a histone demethylase that catalyzes the demethylation of H3K27me3, thereby enhancing gene expression [14]. JMJD3 plays a critical role in macrophage polarization. Liang et al. [15] found that M2-like macrophage polarization is induced by phosphoryl serine produced from dead tumor cells via the PSR-STAT3-JMJD3 axis. Tang et al. [16] found that IL-4 can increase JMJD3 expression and polarize N9 microglia toward M2-like macrophages.

The signal transducer and activator of transcription 6 (STAT6) can be activated by cytokines such as IL-4 and IL-13 [17]. Following phosphorylation, STAT6 translocates to the nucleus to activate transcription of genes associated with M2-like macrophages [18]. Interferon regulatory factor 4 (IRF4), a member of the interferon regulatory factor family, is essential for M2-like macrophages polarization [19]. Satoh et al. [20] found that JMJD3 demethylate H3K37me3 at the IRF4 promoter, promoting the expression of IRF4, and polarizing macrophages toward the M2-like type. However, the precise mechanisms by how JMJD3, STAT6, and IRF4 modulate M2-like macrophage polarization remain unclear.

This study demonstrates that a high M2-like macrophages score in breast cancer is associated with poor prognosis and is an independent risk factor for overall survival. IL4 and IL13 promote the polarization of THP-1 cells into M2-like macrophages and the growth of breast cancer cells by activating the STAT6/JMJD3/IRF4 signaling pathway.

## Materials and methods

### The cancer genome atlas cohort analysis

Clinical and RNA sequencing data for 1,095 breast cancer patients were obtained from The Cancer Genome Atlas Breast Invasive Carcinoma (TCGA-BRCA) database (https://portal.gdc.cancer.gov/). Only patients with complete survival information were included in this study. The CIBERSORT algorithm was employed to estimate the relative proportions of infiltrating immune cell types in each sample [21]. Patients were stratified into high and low score groups based on the median value of macrophage M2 score. Differences in clinical characteristics between the two groups were analyzed using the TableOne R package. The association between the macrophage M2 score and overall survival was assessed using the Cox proportional hazards regression model.

All patient data used in this study were obtained from the TCGA database where ethical approval was obtained for all participants. Therefore, there are no ethical issues or conflicts of interest related to this research.

## Cell culture and reagents

The MDA-MB-231, MCF-7 and THP-1 cell lines were obtained from Fuheng Cell Center (Shanghai, China). MDA-MB-231 and MCF-7 cells were maintained in high-glucose Dulbecco's Modified Eagle's Medium (Bio-Channel, Nanjing, China) supplemented with 10% (v/v) fetal bovine serum (Bio-Channel). THP-1 cells were cultured in RPMI 1640 Medium (Bio-Channel), also supplemented with 10% (v/v) fetal bovine serum (Bio-Channel). All cells were grown at 37°C in 5% $CO_2$.

## Macrophage activation and conditioned media (CM) collection

THP-1 cells were seeded in a 6-well plate at a density of $1 \times 10^5$ cells per well and exposed to 50 ng/ml Phorbol 12-myristate 13-acetate (PMA) (P1585; Sigma, St. Louis, MO, USA) for 24 hours. Subsequently, the cells were treated with 20 ng/ml interleukin-4 (IL-4) (200-04-02; PeproTech, Cranbury, NJ, USA) and 20 ng/ml interleukin-13 (IL-13) (HY-P70568; MCE, Monmouth Junction, NJ, USA). AS1517499 (HY-100614, MCE) is a novel and potent STAT6 inhibitor. To investigate the role of STAT6 in M2-like macrophage polarization, AS1517499 (1μM) was administered 30 minutes prior to IL-4 and IL-13 treatment. After 48 hours of incubation, the medium was replaced with RPMI 1640 medium without FBS and the cells were cultured for an additional 24 hours. The medium was then collected, centrifuged, and the supernatants were preserved for further studies.

## Lentiviruses infection

Lentiviruses targeting JMJD3 were provided by GeneChem Technologies (Shanghai, China). The sequences targeting JMJD3 were as follows: #1 GCATCTATCTGGAGAGCAA, #2 GCAAGTGTGGAACTTGCTA, and #3 GGATGGA-GAGATCTTAGAA. The negative control sequence was TTCTCCGAACGTGTCACGT. THP-1 cells ($1 \times 10^5$ per well) were seeded in a six-well plate and infected with sh-JMJD3 and negative control lentiviruses at a multiplicity of infection (MOI) of 100. The volume of lentivirus added was adjusted according to the viral titer, MOI, and cell number. Additionally, 40 μl of HitransG A infection enhancer (provided by GeneChem) was included in the infection mixture.

For JMJD3 overexpression, lentiviruses were supplied by Obio Technology (Shanghai, China). THP-1 cells were infected with these lentiviruses at an MOI of 100, in the presence of 5 μg/ml polybrene-plus (provided by Obio Technology). The medium was replaced 8 hours post-infection. To establish stable recombinant cell lines, 2 μg/ml puromycin (HY-B1743A, MCE) was added to the medium 48 hours after transduction.

## Cell viability assay

A total of 50 μl of MDA-MB-231 cell suspension (containing 2000 cells) was seeded into each well of a 96-well plate. Following this, 50 μl of activated macrophage conditioned media (CM) was added to each well, and the cells were incubated for 48 hours. Subsequently, 10 μl of tetrazolium salt WST-8 (Cell Counting Kit-8 [CCK-8]; HY-K0301; MCE) was added to each well (final volume ratio as 10%). After 1 hour of incubation with WST-8, the absorbance (OD value) was measured at 450 nm using a spectrometer (Bio-Rad, Fremont, CA, USA).

## Cell apoptosis assay

Cell apoptosis was analyzed using Annexin V-FITC and propidium iodide (PI) (556547, BD Pharmingen, San Diego, USA) staining. Briefly, breast cancer cells were cocultured with activated macrophage conditioned medium (CM) for 48 hours. Then they were trypsinized and rinsed with PBS, and resuspended in a binding buffer containing PI and Annexin V-FITC for 20 minutes. The sample were analyzed using a Navios flow cytometer (Beckman Coulter, USA).

## Quantitative reverse transcription polymerase chain reaction (qRT-PCR)

Total RNA was isolated from the cells using Invitrogen TRIzol® reagent (15596026; Waltham, MA, USA). RNA was then converted to complementary DNA (cDNA) using the PrimeScript™ RT Reagent Kit with gDNA Eraser (RR047A, Takara

Biotechnology). Quantitative real-time PCR was performed using TB Green® Premix Ex Taq™ II (Tli RNase H Plus) (RR820A, Takara Biotechnology) on an Applied Biosystems 7500 Detection System. The qPCR primer sequences were as follows: JMJD3 forward, 5'-CTGGAGAGCAAACGGGATG-3' and reverse 5'-AGGGTCTTGGTGGAGAAGAGG-3'; IRF4 forward, 5'- AGGATGGAGCTGACTACGGAACTG-3' and reverse 5'-TTGGCACGAGGAAGCATAGAACAG-3'; Arg1 forward, 5'-GGACCTGCCCTTTGCTGACATC-3' and reverse 5'-TCTTCTTGACTTCTGCCACCTTGC-3'; GAPDH forward, 5'- GGAAGGAAATGAATGGGCAGC-3' and reverse, 5'-CAGGGTTAGTCACCGGCAG-3'. Relative gene expression levels were calculated using the ΔΔCt method with GAPDH as the internal control.

## Western blot

Cell lysates were prepared using cold RIPA lysis buffer (P0013B; Beyotime, Jiangsu, China) supplemented with protease inhibitor cocktail (HY-K0010; MCE), phosphatase inhibitor cocktail (HY-K0021; MCE), and PMSF. Protein concentration was determined using the BCA Protein Assay Kit (P0012; Beyotime Biotechnology, Shanghai, China) according to the manufacturer's instructions. Proteins (20 μg per sample) were separated by SDS-PAGE (Whatman, Maidstone, Kent, UK) and transferred to a PVDF membrane. After blocking non-specific antigens with quick blocking buffer (HY-K1027; MCE), the membranes were incubated overnight at 4°C with primary antibodies: Anti-JMJD3 (1:2000) (PA5−72751; Thermo Fisher Scientific, New York, USA), Anti-Histone H3 (tri methl K27) (1:1000) (ab6002; Abcam, Cambridge, UK), Anti-IRF4 (1:1000) (ab133590; Abcam), Anti-CD206 (1:1000) (MA5−32498; Thermo Fisher Scientific), Anti-STAT6 (phospho Y641) (1:2000) (ab28829; Abcam), Anti-Arg1 (1:500) (14-9779-82; Thermo Fisher Scientific), GAPDH (1:5000) (R1108-1; HUABIO, Hangzhou, China), β -actin (1:10000) (ET1702−52; HUABIO). ECL-Plus (Thermo Fisher Scientific, New York, USA) and secondary HRP-conjugated antibodies were used to visualize proteins. GAPDH and β-actin served as loading controls. Protein expression levels were quantified using Image J software.

## Cytokine detection

Cytokine levels of TNF and IL-10 in the cell supernatants were measured using a human Th1/Th2 cytokine bead array (CBA) kit (RAISECARE, Qingdao, China). Data were collected using a Navios flow cytometer (Beckman Coulter, USA) and analysed with LegendPlex software (VigeneTech, MA, USA).

## Statistical analysis

The correlation between the M2 macrophage score and clinical characteristics was analyzed using Fisher's Exact Test. Cumulative survival was assessed using the Kaplan-Meier method. The associations of clinical features and the M2 macrophage score with survival were analyzed using the Cox regression model from the survival package. Cell experimental data were presented as mean±standard error. One-way analysis of variance (ANOVA) and the Student's t-test were employed to determine statistical significance, with p-value of less than 0.05 considered significant. Each experiment was conducted with at least three replicates to ensure reliable data. Statistical analyses were performed using R software (v4.4.1) and GraphPad Prism (v10.0).

## Results

### M2 macrophage score correlated with clinicopathological characteristics and prognosis in BRCA

To explore the association between the M2 macrophage score and clinicopathological features, 1,095 BRCA patients from the TCGA database were divided into high and low M2 macrophage score groups based on the median score. Clinical baseline data were summarized in Supplementary S1 Table. Significant differences were observed between the two groups in age, estrogen receptor (ER), and progesterone receptor (PR) status ($p < 0.05$). Subgroup analyses further demonstrated that higher M2 macrophage scores were significantly enriched in patients aged over 60 years, those with stage III or IV disease, and in ER-positive and PR-positive subgroups ($p < 0.05$; Fig 1A).

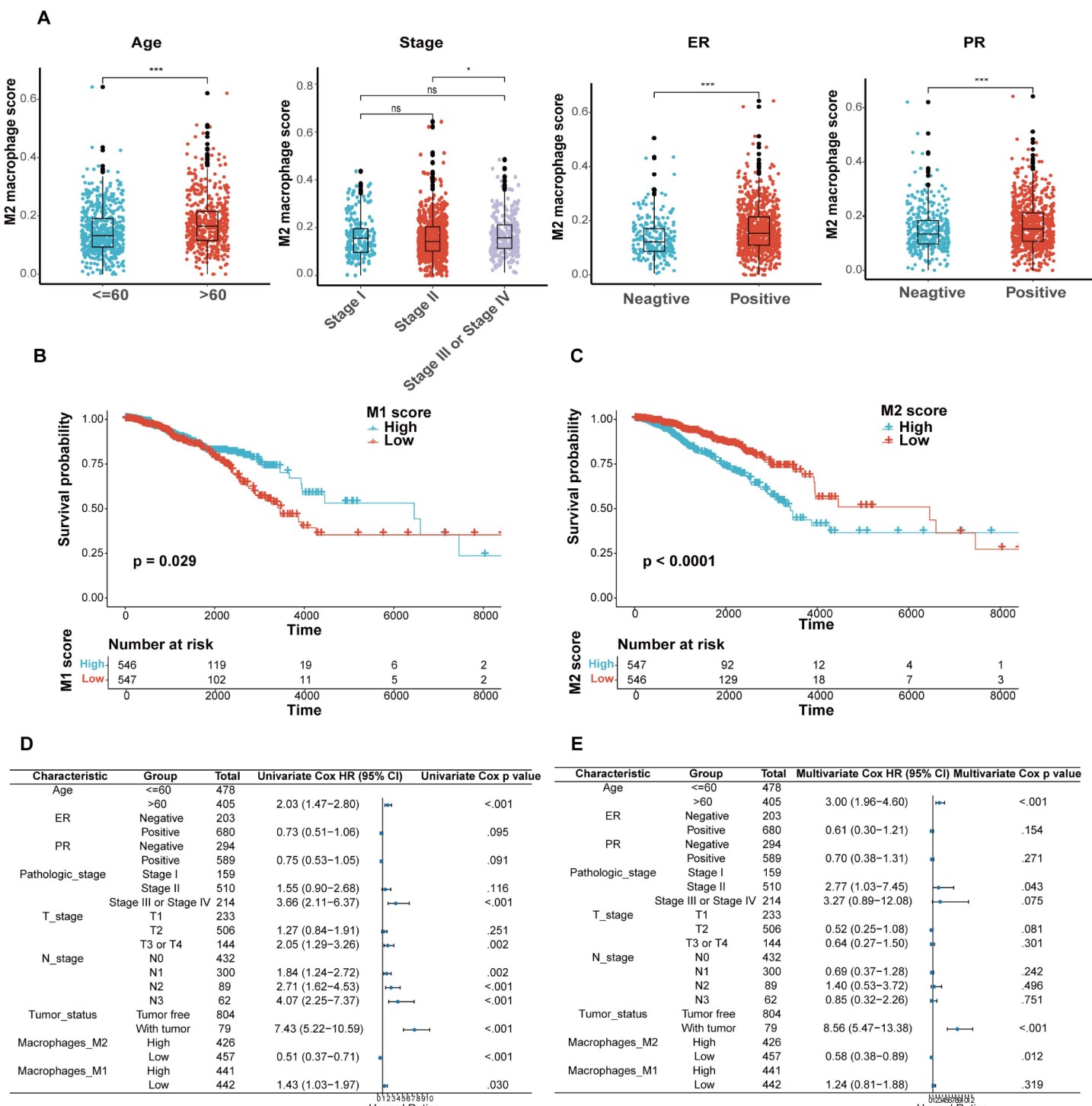

**Fig 1. Correlation of M2 macrophage score with different clinicopathological characteristics and the prognosis in TCGA-BRCA. (A)** Correlation analysis under different clinical subgroups: age, pathologic stage, ER, and PR expression. Kaplan-Meier survival analysis based on M1 macrophage score (B) and M2 macrophage score **(C)**. Forest plot displaying the hazard ratio (HR) with 95% confidence interval (CI) for BRCA patients based on univariate Cox analysis (D) and multivariate Cox analysis **(E)**. * p < 0.05; ** p < 0.01; *** p < 0.001; ns, not significant. Abbreviations: ER, estrogen receptor; PR, progesterone receptor; Stage, pathologic stage.

Kaplan-Meier survival analysis indicated that patients with lower M1 macrophage scores (Fig 1B) or higher M2 macrophage scores (Fig 1C) exhibited significantly poorer overall survival (OS) (p<0.05). Univariate Cox regression analysis identified age (> 60 years, HR=2.03, p<0.001), pathologic stage (III or IV, HR=3.66, p<0.001), pathologic T stage (T3 or T4, HR=2.05, p=0.002), N stage (N1, HR=1.84, p=0.002; N2, HR=2.71, p<0.001; N3, HR=4.07, p<0.001), tumor status (presence of tumor, HR=7.43, p<0.001), low M2 macrophage score (HR=0.51, p<0.001), and low M1 macrophage score (HR=1.43, p=0.030) as factors related to OS in BRCA patients (Fig 1D). Multivariate Cox regression analysis further identified age (> 60 years, HR=3.00, p<0.001), tumor status (presence of tumor, HR=8.56, p<0.001), and low M2 macrophage score (HR=0.58, p=0.012) as independent risk factors for OS in BRCA patients (Fig 1E).

**IL-4 and IL-13 induced M2-like macrophage polarization and upregulated the expression of JMJD3**

To explore the mechanism underlying M2-like macrophage polarization, we treated THP-1 cells with IL-4 and IL-13 and examined the morphological and gene expression changes. Initially, THP-1 cells were circular, suspended, and uniform in size and shape (Fig 2A, left). After treatment with 50 ng/ml PMA for 24 hours, the cells adhered to the dish and developed pseudopods (Fig 2A, middle). Following an additional 24 hours of treatment with IL-4 and IL-13, the cells adhered more tightly and exhibited significant morphological changes, including irregular and elongated shapes with longer tentacles (Fig 2A, right). RT-PCR and Western blot (WB) analyses were conducted to assess the expression of M2-like macrophage markers Arg1 (Fig 2B, 2C). The cytokine levels of TNF, and IL-10 in the macrophage supernatant were determined using flow cytometer (Fig 2D). The findings indicated that IL-4 and IL-13 effectively upregulated the M2-like macrophage markers Arg1 and IL10, while downregulated the M1-like macrophage marker TNF, demonstrating the polarization of THP-1 cells to M2-like macrophages phenotype.

Previous studies have indicated that the polarization of M2-like macrophages is regulated by the JMJD3 and STAT6/IRF4 signaling pathways [19,20,22,23]. To further investigate this possibility, we examined the expression of JMJD3, p-STAT6-Y641, and IRF4 in the macrophage-like cells. Our results showed a significant increase in both JMJD3 and IRF4 mRNA expression, as well as elevated protein levels of JMJD3, p-STAT6-Y641, and IRF4 in M2-like macrophages (Fig 2E, 2F). These findings suggested that JMJD3, IRF4, and p-STAT6-Y641 were activated during the polarization of macrophages toward the M2-like phenotype.

**JMJD3 promoted M2-like polarization in macrophages**

To investigate the role of JMJD3 in M2-like macrophage polarization, we employed lentiviruses carrying shJMJD3 and oeJMJD3 to suppress and overexpress JMJD3 in THP-1 cells, respectively. JMJD3 expression was significantly reduced with shJMJD3 and increased with oeJMJD3 lentiviruses infection compared to the control (Fig 3A). Consistent with its known demethylase activity, knockdown of JMJD3 markedly elevated H3K27me3 accumulation (S1A Fig), whereas JMJD3 overexpression led to a significant reduction in H3K27me3 levels (S1B Fig).

Following lentiviruses infection, THP-1 cells were treated with PMA alone or with IL-4 and IL-13. JMJD3 knockdown resulted in a significant decrease in Arg1 mRNA (Fig 3B) and CD206 protein levels (Fig 3E), accompanied by an increase in TNF concentration compared to controls (Fig 3C). Conversely, JMJD3 overexpression resulted in increased Arg1(Fig 3B) and CD206 expression (Fig 3E) and decreased TNF concentrations (Fig 3C). These results suggest that JMJD3 is crucial for M2-like polarization in THP-1 cells.

Additionally, IRF4 expression was significantly lower with JMJD3 knockdown and higher with JMJD3 overexpression compared to control group (Fig 3D, 3E), indicating that JMJD3 modulates IRF4 expression in M2-like macrophages.

**JMJD3 promotes M2-like macrophage polarization through the STAT6/IRF4 axis**

Given that IL-4 and IL-13 stimulation increased the expression of JMJD3, p-STAT6, and IRF4 in macrophages, we hypothesized that IL-4 and IL-13 upregulated JMJD3 and IRF4 via STAT6 activation. To test this hypothesis, we utilized

 

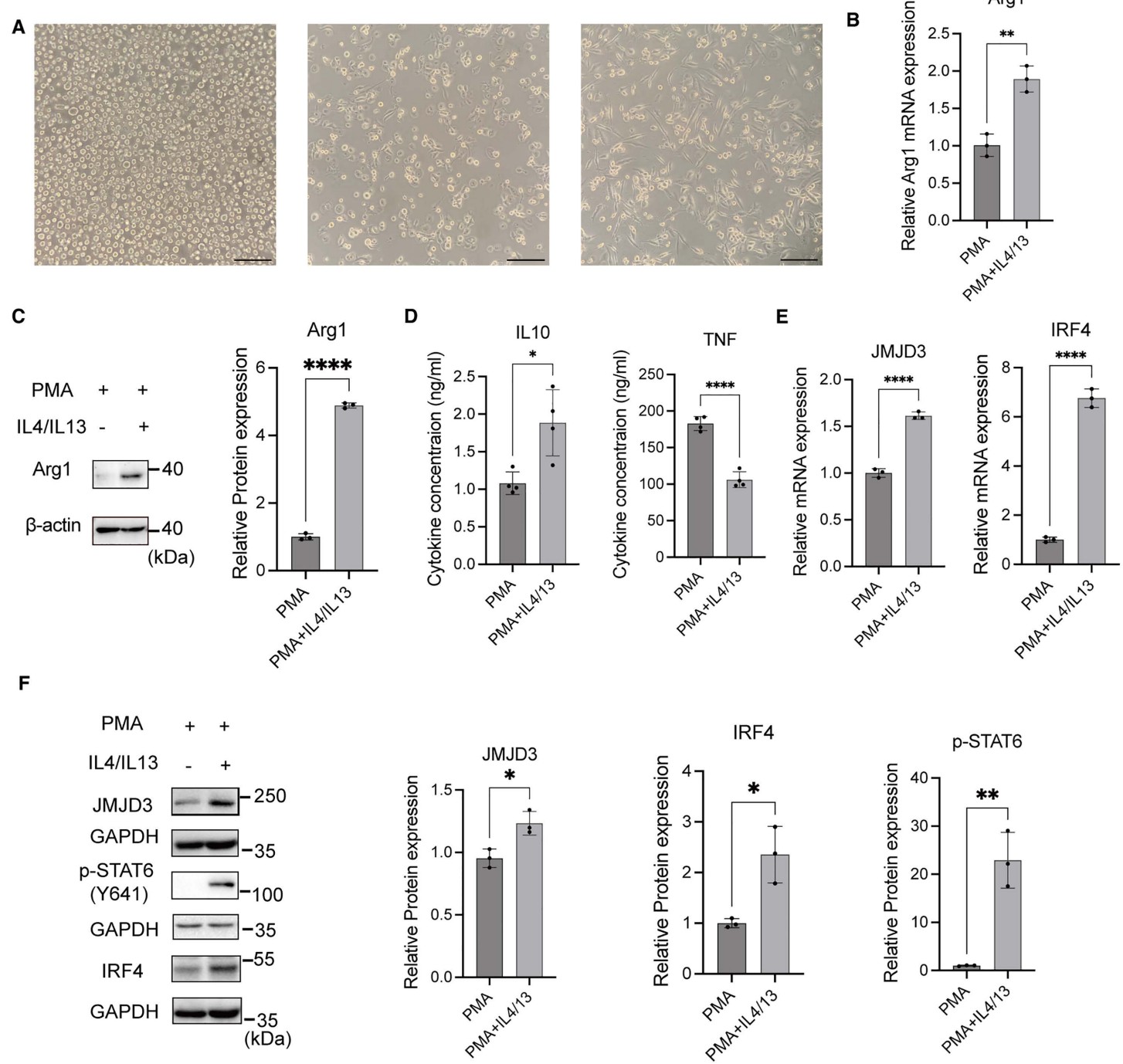

**Fig 2. IL-4 and IL-13 induce M2-like macrophage polarization and upregulate JMJD3 expression.** THP-1 cells were treated under different conditions: PMA (50 ng/ml) alone, or PMA (50 ng/ml) for 24 hours, followed by a medium change and subsequent treatment with IL-4 (20 ng/ml) and IL-13 (20 ng/ml) for 48 hours. After this, the medium was replaced with RPMI 1640 medium without FBS for 24 hours. Protein and mRNA were then extracted, and the supernatant was collected. **(A)** Morphological features of THP-1 cells under different conditions. Left: untreated THP-1 cells. Middle: PMA (50 ng/ml) for 24 hours. Right: PMA for 24 hours, followed by IL-4 and IL-13 (20 ng/ml) and IL-13 (20 ng/ml) for 48 hours. Scale bar: 50 μm. **(B)** Relative expression of Arg1 mRNA. **(C)** Relative expression of Arg1 protein. **(D)** Concentrations of TNF and IL-10 in the supernatant. **(E)** Relative mRNA expression of JMJD3 and IRF4. **(F)** Protein expression of JMJD3, p-STAT6-Y641, and IRF4. The experiments were replicated three times. * $p < 0.05$, ** $p < 0.01$, ***$p < 0.001$, ****$p < 0.0001$. Abbreviations: Arg1, arginase 1; PMA, Phorbol 12-myristate 13-acetate; IL-10, interleukin-10; TNF, tumor necrosis factor; JMJD3, Jumonji domain-containing protein 3; IRF4, interferon regulatory factor 4; p-STAT6, phosphorylated signal transducer and activator of transcription 6.

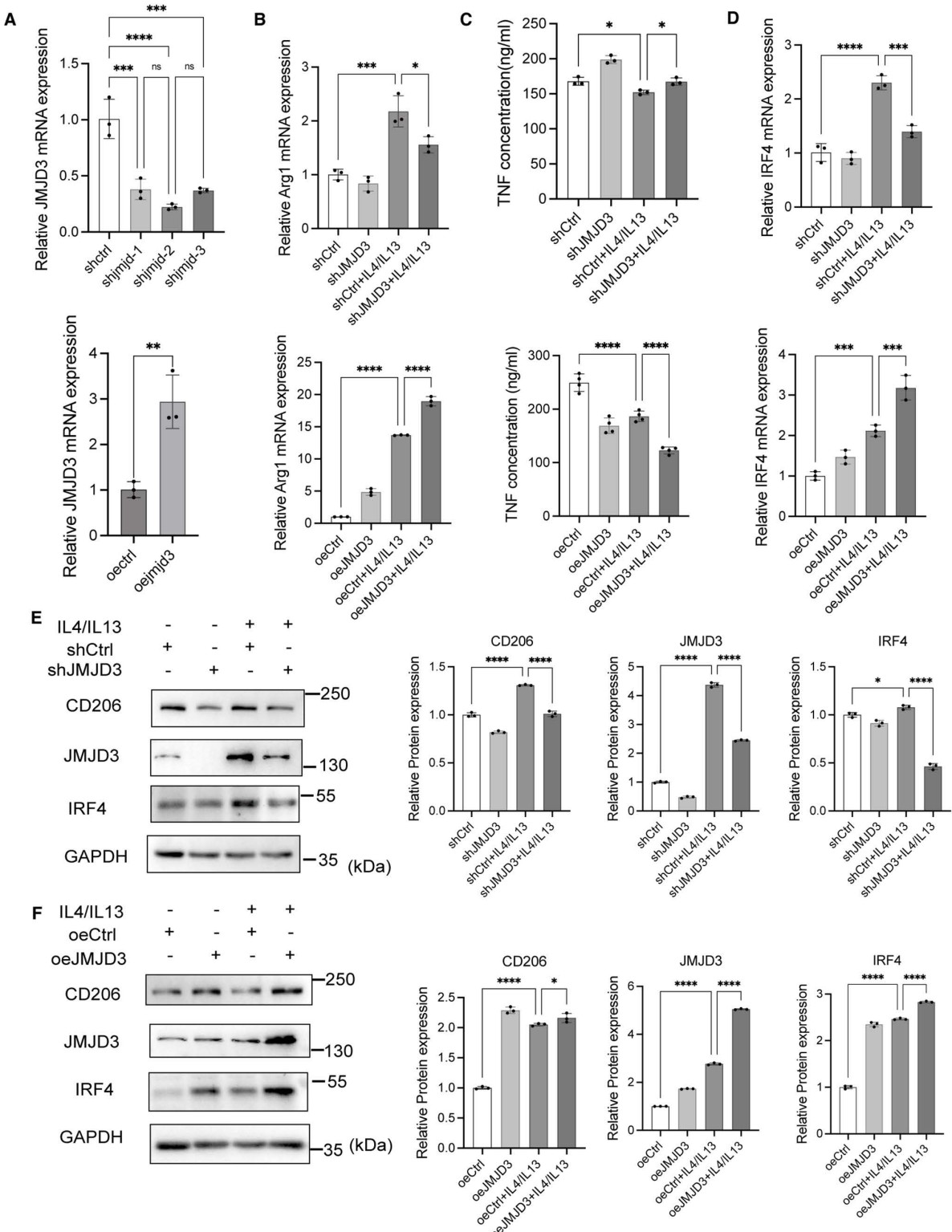

**Fig 3. JMJD3 promotes M2-like polarization in macrophages. (A)** THP-1 cells were infected with shJMJD3 or oeJMJD3 lentiviruses for 72 hours. JMJD3 mRNA expression was then assessed. shCtrl and oeCtrl: negative control lentiviruses. shJMJD3: THP-1 cells infected with lentiviruses to knock-down JMJD3. oeJMJD3: THP-1 cells infected with lentiviruses to overexpress JMJD3. Following lentivirus infection, THP-1 cells were treated with PMA

for 24 hours, followed by IL-4 and IL-13 for 48 hours. After this, the medium was replaced by RPMI 1640 medium without FBS for an additional 24 hours. Protein and mRNA were then extracted, and the supernatant was collected. **(B)** Relative expression of Arg1 mRNA. **(C)** Concentrations of TNF in the supernatant of different groups of macrophages. **(D)** Relative expression of IRF4 mRNA. **(E, F)** Western blot analysis of CD206, JMJD3, and IRF4 protein. The experiments were replicated three times. * p < 0.05, ** p < 0.01, ***p < 0.001, ****p < 0.0001. Abbreviations: JMJD3, Jumonji domain-containing protein 3; Arg1, arginase 1; TNF, tumor necrosis factor; IRF4, interferon regulatory factor 4; CD206, cluster of differentiation 206.

AS1517499 to inhibit STAT6 activity and examined its effects on M2-like macrophage polarization as well as JMJD3 and IRF4 expression. Treatment with AS1517499 resulted in a decrease in Arg1 mRNA expression (Fig 4A), and the concentrations of IL-10 (Fig 4C), while increasing the concentration of TNF (Fig 4B). These results suggest that AS1517499 impedes macrophage polarization toward the M2-like phenotype.

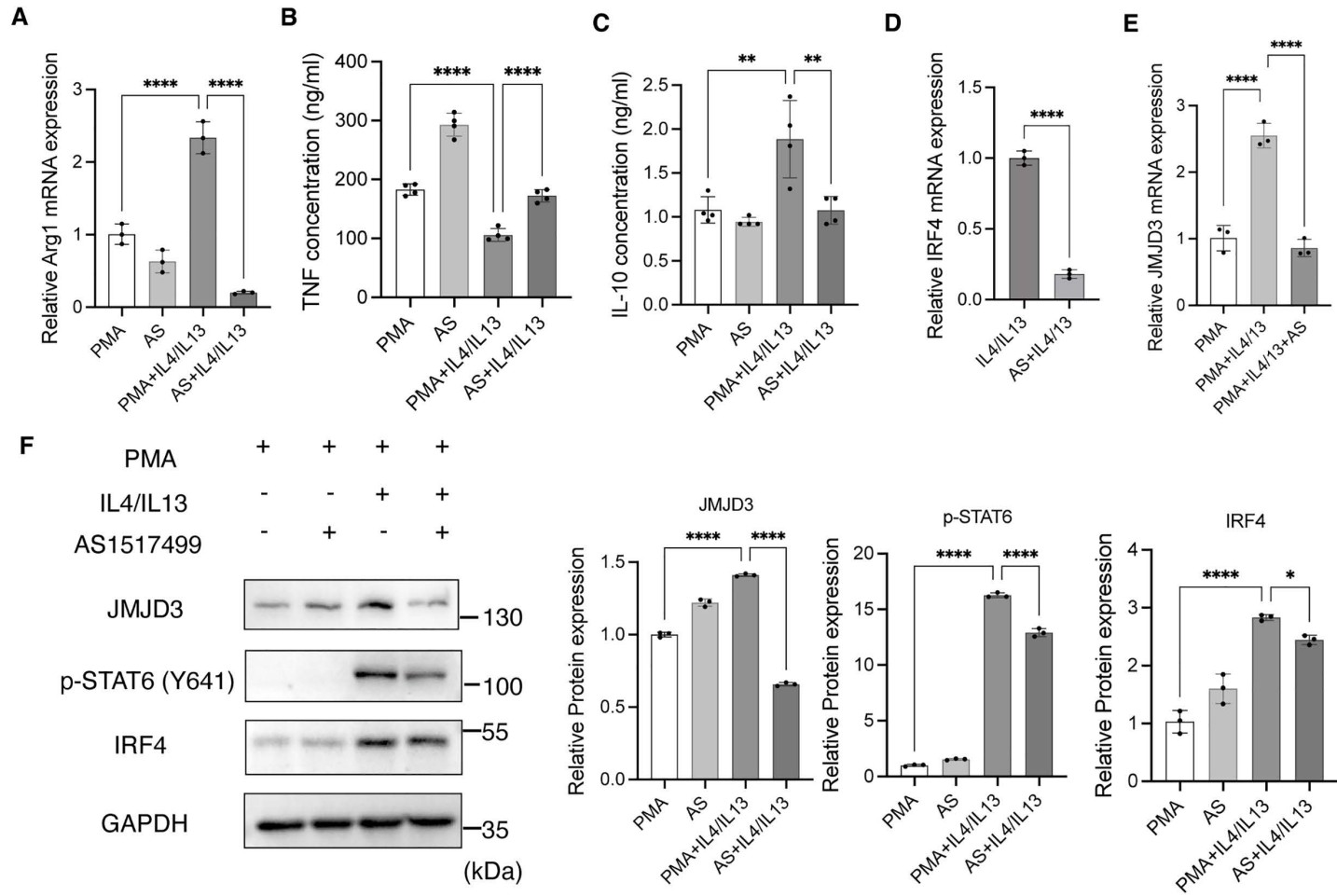

**Fig 4. JMJD3 promotes the M2-like macrophage polarization through the STAT6/IRF4 axis.** THP-1 cells were first treated with PMA for 24 hours. Following this, AS1517499 (1μM) was administered for 30 minutes, and then the cells were treated with IL-4 and IL-13. 48 hours later, the medium was replaced with RPMI 1640 medium without FBS and cells were incubated for an additional 24 hours. Protein and mRNA were then extracted, and the supernatant was collected. **(A)** The mRNA expression of Arg1. **(B)** Concentrations of TNF in the supernatant of different groups of macrophages. **(C)** Concentrations of IL-10 in the supernatant of different groups of macrophages. **(D)** Expression of IRF4 mRNA. **(E)** Expression of JMJD3 mRNA. **(F)** Western blot analysis of JMJD3, p-STAT6, and IRF4 protein expression. The experiments were replicated three times. * p < 0.05, ** p < 0.01, ***p < 0.001, ****p < 0.0001. Abbreviations: JMJD3, Jumonji domain-containing protein 3; Arg1, arginase 1; TNF, tumor necrosis factor; IRF4, interferon regulatory factor 4; IL-10, interleukin-10; p-STAT6, phosphorylated signal transducer and activator of transcription 6.

Further analysis revealed that AS1517499 significantly decreased the expression of p-STAT6, JMJD3, and IRF4 at both the mRNA (Fig 4D, 4E) and protein levels (Fig 4F). These findings suggest that STAT6 regulates JMJD3 and IRF4 expression in M2-like macrophages and that AS1517499 may potentially modulate M2-like macrophage-mediated immune responses.

## The M2-like macrophage regulated by JMJD3 and STAT6 promoted the growth of breast cancer cells

To investigate the impact of M2-like macrophages on breast cancer cell proliferation and apoptosis, MDA-MB-231 and MCF-7 cells were cultured either alone or with conditioned medium from macrophages for 48 hours. Cell proliferation was measured using the CCK8 assay, while apoptosis was assessed by flow cytometry with Annexin V and PI staining.

Results revealed that MDA-MB-231 cells cultured with conditioned medium from IL-4 and IL-13-treated macrophages showed increased proliferation (Fig 5A) and decreased apoptosis (Fig 5D) compared to those cultured alone or with PMA-treated macrophage medium. Conversely, MDA-MB-231 cells cultured with conditioned medium from macrophages treated with AS1517499 or shJMJD3 lentiviruses significantly reduced cell proliferation (Fig 5A, 5B) and increased cell apoptosis (Fig 5D) compared to controls. On the other hand, MDA-MB-231 cells cultured with conditioned medium from macrophages treated with oeJMJD3 lentiviruses enhanced cell proliferation (Fig 5C) and reduced cell apoptosis (Fig 5D). These findings suggest that JMJD3 and STAT6-regulated M2-like macrophages facilitate the growth of MDA-MB-231 breast cancer cells.

Similarly, MCF-7 cells exhibited consistent trends: conditioned medium from JMJD3-silenced macrophages (shJMJD3) markedly suppressed cell proliferation (S2A Fig) and increased cell apoptosis (S2B Fig), whereas medium from macrophages overexpressing JMJD3 (oeJMJD3) promoted MCF-7 cell proliferation (S2A Fig) and inhibited apoptosis (S2B Fig).

## Discussion

Breast cancer is the most prevalent cancer and the leading cause of cancer-related deaths in women [1]. Macrophages, which are the most abundant cells in the breast cancer microenvironment, play a crucial role in tumor development, growth, treatment response, and prognosis [11–13]. In this study, we demonstrated that M2-like macrophage score is significantly associated with patient age, as well as ER and PR status, and is an independent risk factor of breast cancer prognosis based on immune infiltration analysis of the TCGA-BRCA dataset. Furthermore, we found that M2-like macrophages promote breast cancer cell proliferation, and that IL4 and IL13 induce M2-like macrophage polarization through the STAT6/JMJD3/IRF4 pathway.

To investigate the mechanism of M2-like macrophage polarization and its effect on breast cancer, we used human THP-1 cells as a monocyte model. The cells were treated with PMA, IL-4, and IL-13, resulting in a morphological shift from a circular shape to an irregular or shuttle-shaped form with extended tentacles. This was accompanied by an increased expression of the M2-like macrophage marker Arg1, elevated levels of IL-10 in the supernatant, and a reduction in the M1-like macrophage marker TNF, indicating successful polarization to the M2-like macrophage phenotype. Furthermore, when breast cancer cells were cocultured with the conditioned medium from these M2-like macrophages, they exhibited increased proliferation and reduced apoptosis, suggesting that M2-like macrophages promote breast cancer cell growth and inhibit cell death.

In eukaryotic cells, histones and DNA form the fundamental unit of chromatin known as the nucleosome [24]. The N-terminal tails of histones undergo various post-translational modifications, including methylation, acetylation, and ubiquitination. These modifications influence chromatin structure and gene transcription [25]. Specifically, trimethylation of histone H3 at lysine 27 (H3K27me3) is associated with gene transcription suppression. JMJD3, a histone demethylase, acts on H3K27 sites, catalyzing the demethylation of H3K27me3 and promoting gene transcription [14]. Consistent with this mechanism, our Western blot analysis showed that JMJD3 overexpression markedly decreased global H3K27me3 levels, whereas JMJD3 knockdown led to an accumulation of H3K27me3, confirming that JMJD3 functions as an

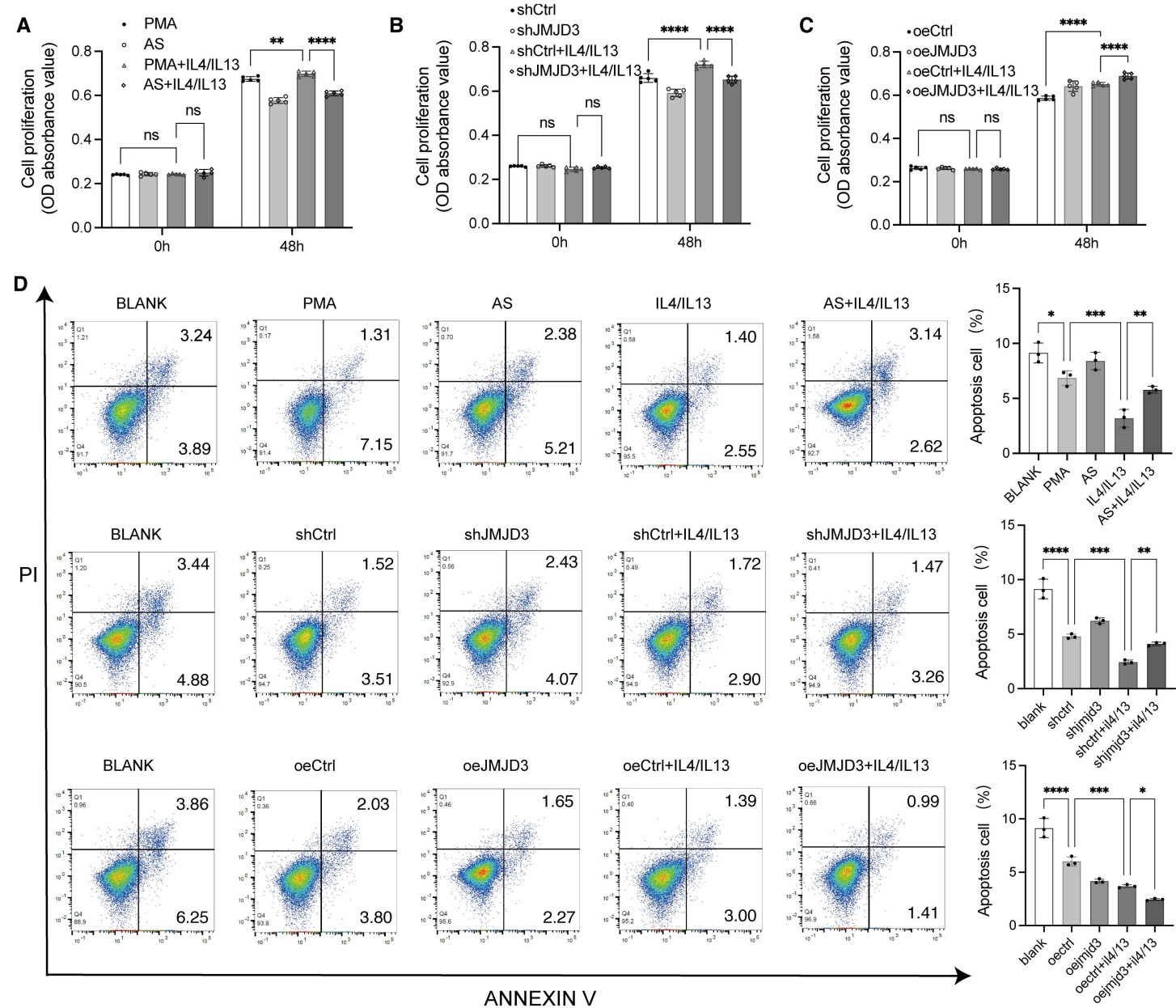

**Fig 5. Effect of JMJD3 and STAT6 regulated M2-like macrophages on the growth of MDA-MB-231 cells.** MDA-MB-231 breast cancer cells were cultured with different types of macrophage-conditioned media for 48 hours. (A, B, **C**) Proliferation of MDA-MB-231 breast cancer cells assessed using the CCK8 assay. **(D)** Apoptosis rates of MDA-MB-231 cells measured by flow cytometer. The experiments were replicated three times. * $p < 0.05$, ** $p < 0.01$, ***$p < 0.001$, ****$p < 0.0001$. Abbreviations: PMA, Phorbol 12-myristate 13-acetate; OD, optical density; PI, propidium iodide.

H3K27me3-specific demethylase in THP-1 cells. JMJD3 plays a crucial role in macrophage polarization. Zhu et al. [26] found that JMJD3 induced M2-like polarization of testicular macrophages, reduced hyperglycemia-induced inflammation in the male reproductive system, and improved the reproductive success of diabetic mice. Raines et al. [27] discovered that IL-4 stimulated serine production and activated the protein kinase RNA-like ER kinase (PERK) signaling cascade in macrophages. This enhanced JMJD3's epigenetic function, leading to demethylation of H3K27me3 on M2-like marker

gene promoters and promoting M2-like macrophage activation and proliferation. Ishii et al. [28] discovered that IL-4 treatment significantly increased the expression of M2-like marker genes and JMJD3 in bone marrow-derived macrophages. In our study, we found that JMJD3 expression increased when IL-4 and IL-13 induced THP-1 cells to differentiate into M2-like macrophages. Knockdown of JMJD3 resulted in reduced expression of the M2-like macrophage marker Arg1 and increased levels of the M1-like macrophage marker TNF. Conversely, overexpression of JMJD3 in THP-1 cells yielded opposite results. These findings highlight the critical role of JMJD3 in M2-like macrophage polarization.

The JAK/STAT pathway, activated by cytokines, plays a crucial role in regulating cell proliferation, apoptosis, and immune responses [29]. STAT6 can be activated by cytokines such as IL-4 and IL-10, leading to phosphorylation of STAT6 and its translocation to the nucleus, where it initates the transcription of M2-like macrophage marker genes [17]. Fu et al. [22] found that IL-4/STAT6 signaling enhances M2-like macrophages polarization, thereby promoting lung cancer progression. IRF4 is involved in various biological processes, including the cell cycle, apoptosis, and inflammation. Satoh et al. [20] demonstrated that JMJD3 facilitates the demethylation of H3K37me3 in the IRF4 promoter, thereby promoting M2-like macrophage polarization. Our study showed that treating THP-1 cells with PMA, IL-4, and IL-13 increased the expression of JMJD3, p-STAT6-Y641, and IRF4. JMJD3 knockdown resulted in decreased IRF4 expression, while JMJD3 overexpression increased IRF4 levels. Inhibition of STAT6 reduced both JMJD3 and IRF4 expression. Co-culturing breast cancer cells with conditioned media from JMJD3 knockdown or STAT6-inhibited macrophages resulted in decreased cancer cell proliferation and increased apoptosis. These findings suggest that JMJD3, p-STAT6-Y641 and IRF4 are activated during M2-like macrophage polarization. JMJD3 regulated IRF4 expression, and macrophages regulated by JMJD3 and STAT6 impact breast cancer progression. However, the specific regulatory mechanisms need further studies.

However, there are some limitations in this study. First, although we demonstrated that M2-like macrophages regulated by JMJD3 and STAT6 promote breast cancer cell proliferation and inhibit apoptosis, the specific macrophage-derived factors responsible for mediating these effects remain unclear. Future studies will employ cytokine profiling, RNA sequencing, or Olink proteomics panels to identify the soluble molecules secreted by JMJD3-activated macrophages that drive tumor growth. Second, while our data support that JMJD3 regulates IRF4 expression and contributes to M2-like polarization, the direct molecular mechanism linking JMJD3 to the transcriptional activation of IRF4 and other M2-associated genes has not been fully elucidated. Planned experiments, including chromatin immunoprecipitation (ChIP)-qPCR and promoter luciferase reporter assays, will be conducted to clarify whether JMJD3 directly binds or epigenetically modifies the IRF4 promoter. Third, our conclusions are based primarily on in vitro cell culture systems. The lack of in vivo validation limits the generalizability of our findings. Future studies using mouse models of breast cancer or macrophage-specific JMJD3 knockout mice will be essential to confirm the role of the JMJD3/STAT6/IRF4 axis in tumor progression and to explore its potential as a therapeutic target.

## Conclusions

In summary, IL-4 and IL-13 induce M2-like macrophage polarization through the STAT6/JMJD3/IRF4 pathway. Furthermore, conditioned media from JMJD3-regulated macrophages promote breast cancer cell growth and inhibit apoptosis (Fig 6), highlighting the role of macrophage-derived factors in the tumor microenvironment. Further studies are needed to identify the specific factors and validate these effects in vivo.

## Supporting information

**S1 Table. The relationship between the clinicopathological characteristics and M2 macrophage score in BRCA.** (DOCX)

**S1 Fig. JMJD3 regulated global H3K27me3 levels.** Western blot analysis showing that JMJD3 knockdown elevated H3K27me3 accumulation (A), whereas JMJD3 overexpression reduced H3K27me3 (B) levels in THP-1 cells. GAPDH was used as a loading control. Significance levels: **, $p < 0.01$, ****, $p < 0.001$. (TIF)

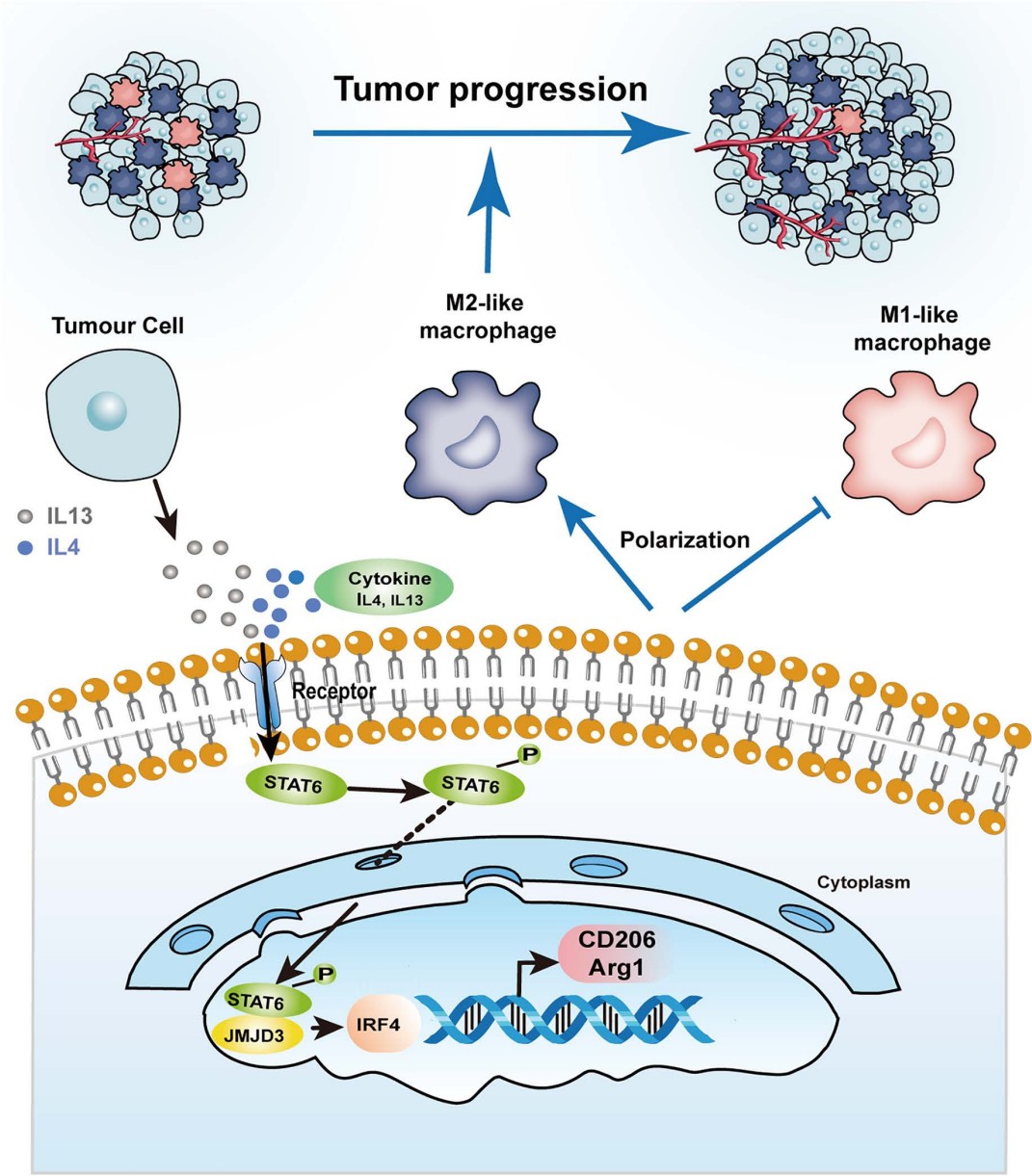

**Fig 6. Schematic illustration of the regulatory mechanism by which JMJD3 influences M2-like macrophage polarization and promotes breast cancer cell growth through the STAT6/IRF4 axis.**

**S2 Fig. M2-like macrophages regulated by JMJD3 promote the growth of MDA-MB-231 breast cancer cells.** MCF-7 breast cancer cells were cultured with different types of macrophage-conditioned media for 48 hours. A. Proliferation of MCF-7 breast cancer cells assessed using the CCK8 assay. D. Apoptosis rates of MCF-7 cells measured by flow cytometer. Error bars represent mean±SD. Significance levels: * p<0.05, ** p<0.01, ***p<0.001, ****p<0.0001. (TIF)

**S1 File. S1 raw images.**
(PDF)

## Author contributions

**Conceptualization:** Juan Lyu, Lihong Zhang.

**Data curation:** Lihong Zhang.

**Formal analysis:** Juan Lyu.

**Funding acquisition:** Juan Lyu, Ying Qian.

**Investigation:** Juan Lyu, Enqin Wang, Hongkun Xu.

**Methodology:** Juan Lyu, Shanmei Lyu, Ying Qian, Xiuping Xu.

**Resources:** Liangfeng Hu, Lihong Zhang.

**Supervision:** Lihong Zhang.

**Validation:** Fen Ye, Qing Wang, Tao Lu.

**Visualization:** Shanmei Lyu.

**Writing – original draft:** Juan Lyu.

**Writing – review & editing:** Lihong Zhang.

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
