## [Decision Letter · Decision Letter 0]

25 Sep 2025

Dear Dr. Zhang,

Thank you for submitting your manuscript to PLOS ONE. After careful consideration, we feel that it has merit but does not fully meet PLOS ONE’s publication criteria as it currently stands. Therefore, we invite you to submit a revised version of the manuscript that addresses the points raised during the review process.

We look forward to receiving your revised manuscript.

Kind regards,

Li Yang, M.D.

Academic Editor

PLOS ONE

Journal Requirements:

https://journals.plos.org/plosone/s/file?id=ba62/PLOSOne_formatting_sample_title_authors_affiliations.pdf ..

“The study was supported by the Zhejiang Provincial Natural Science Foundation of China under Grand No. LQ24H200001, Medical Science and Technology Project of Zhejiang Provincial Health Commission Grand No.2022RC273, 2022RC275 2023KY353, 2024KY462, and by Shaoxing City Health Science and Technology Plan Grand No. 2022SY020.”

“The study was supported by the Zhejiang Provincial Natural Science Foundation of China under Grand No. LQ24H200001, Medical Science and Technology Project of Zhejiang Provincial Health Commission Grand No.2022RC273, 2022RC275 2023KY353, 2024KY462, and by Shaoxing City Health Science and Technology Plan Grand No. 2022SY020.”

4. In the online submission form you indicate that your data is not available for proprietary reasons and have provided a contact point for accessing this data. Please note that your current contact point is a co-author on this manuscript. According to our Data Policy, the contact point must not be an author on the manuscript and must be an institutional contact, ideally not an individual. Please revise your data statement to a non-author institutional point of contact, such as a data access or ethics committee, and send this to us via return email. Please also include contact information for the third party organization, and please include the full citation of where the data can be found.

“The study was supported by the Zhejiang Provincial Natural Science Foundation of China under Grand No. LQ24H200001, Medical Science and Technology Project of Zhejiang Provincial Health Commission Grand No.2022RC273, 2022RC275 2023KY353, 2024KY462, and by Shaoxing City Health Science and Technology Plan Grand No. 2022SY020.”

“The study was supported by the Zhejiang Provincial Natural Science Foundation of China under Grand No. LQ24H200001, Medical Science and Technology Project of Zhejiang Provincial Health Commission Grand No.2022RC273, 2022RC275 2023KY353, 2024KY462, and by Shaoxing City Health Science and Technology Plan Grand No. 2022SY020.”

7. PLOS ONE now requires that authors provide the original uncropped and unadjusted images underlying all blot or gel results reported in a submission’s figures or Supporting Information files. This policy and the journal’s other requirements for blot/gel reporting and figure preparation are described in detail at https://journals.plos.org/plosone/s/figures#loc-blot-and-gel-reporting-requirements and https://journals.plos.org/plosone/s/figures#loc-preparing-figures-from-image-files. When you submit your revised manuscript, please ensure that your figures adhere fully to these guidelines and provide the original underlying images for all blot or gel data reported in your submission. See the following link for instructions on providing the original image data: https://journals.plos.org/plosone/s/figures#loc-original-images-for-blots-and-gels.

Additional Editor Comments:

Thanks for submitting your work to PLOS ONE. Your manuscript has now been assessed by our editorial team and external peer experts. While they found it interesting, you will see that they have raised many serious problems and are advising that you revise your manuscript thoroughly. At the same time, please submit the point-by-point responses to reviewers' comments. If you are prepared to undertake the work required, I would be pleased to reconsider my decision. Please note that this revision decision does not assure the acceptance of your work. Thanks for the opportunity to consider your work.

Reviewers' comments:

Reviewer's Responses to Questions

**Comments to the Author**

1. Is the manuscript technically sound, and do the data support the conclusions?

Reviewer #1: Yes

Reviewer #2: Partly

2. Has the statistical analysis been performed appropriately and rigorously?

Reviewer #1: Yes

Reviewer #2: Yes

3. Have the authors made all data underlying the findings in their manuscript fully available?

Reviewer #1: Yes

Reviewer #2: Yes

4. Is the manuscript presented in an intelligible fashion and written in standard English?

Reviewer #1: Yes

Reviewer #2: Yes

Reviewer #1: The research article “JMJD3 regulates the M2 macrophage polarization and promotes the growth of breast cancer cells via STAT6/IRF4 axis” by Juan Lyu et al investigates the role of JMJD3 in promoting M2 macrophage polarization through the STAT6/IRF4 signaling axis and draws a correlation between breast cancer progression and M2 TAMs.

Using data from over 1,000 breast cancer patients from the Cancer Genome Atlas (TCGA), the authors found that high levels of M2 macrophages correlate with advanced clinical features (e.g., age >60, ER+/PR+ status, late-stage tumors) and poorer overall survival.

In vitro, the authors induced M2 macrophage-like polarization by exposing THP-1 monocytes to PMA, IL-4, and IL-13. These M2 macrophage-like cells showed increased expression of JMJD3, IRF4, and phosphorylated STAT6. Knockdown of JMJD3 or inhibition of STAT6 reduced the expression of IRF4 and M2 markers while overexpression of JMJD3 enhanced M2 polarization and further supported cancer cell growth.

Mechanistically, the study demonstrates that JMJD3 promotes M2 polarization through the STAT6/IRF4 pathway. STAT6 activation triggered by IL-4/IL-13 increases JMJD3 expression, which in turn enhances IRF4 transcription. Disruption of this axis using a STAT6 inhibitor AS1517499 or JMJD3 knockdown resulted in impaired M2 polarization. In turn, MDA-MB cancer cells exposed to conditioned media of polarized THP-1 cells whose STAT6/JMJD3/IRF4 pathway had been disrupted showed decreased proliferation and increased apoptosis.

MINOR REVISIONS

THP-1 is monocytic cell line that can be differentiated into macrophage-like cells. The authors should refrain from classifying differentiated cells as macrophages and instead classify them as macrophage-like. Point in case, macrophages do not typically produce IL-2. Therefore, the inclusion of IL-2 measurement in figure 2D and referring to IL-2 as an M2 marker is incorrect and should be revised. Typically, M2 macrophages are characterized by IL-10, IL-4, and IL-13 secretion.

Line 199/200: please provide a reference for the statement.

How do the authors account for the practically neglectable differences in apoptosis of MDA-MB-231 cells when subject conditioned medium from polarized THP-1/macrophages overexpressing or lacking JMJD3?

The underlying concept of the paper is the role of JMJD3 in catalyzing the demethylation of H3K27me3 and promoting gene transcription. However, at no point in their model of silencing or overexpressing JMJD3 the authors actually show the effects on H3K27me3 demethylation. The authors should consider doing so as it would further validate their observations and strengthen the paper.

In the proposed model (Figure 6) the authors include CD163. However, at no point in the paper CD163 is mentioned. As such it should be removed from the model. Furthermore, the inclusion of “Cytokine” is not helpful when the authors know the cytokines in question: IL4 and IL13. Please substitute cytokine by IL4 and IL13 and make sure to use different colors for the dots, so that it is easily understood that there are two cytokines involved. I also find that the blue arrows that indicate what processes are occurring are not very intuitive. The authors might want to consider improving that.

Finally, the authors need to provide data points for figures 2B, 2F, 3H, 3J, 3L, 4C, 4G, 5A, 5B, and 5C. Data points should also be more visible in all the figures. Additionally, the authors need to provide information on how many times the essays were done, if we are looking at biological repeats or technical repeats.

Reviewer #2: 1. TCGA analysis showed higher M2 scores in ER-positive and PR-positive patients. However, the functional assay was performed on the MDA-MB-231 cell line (ER-negative, PR-negative). Would the authors could explain about this? Does the STAT6/JMJD3/IRF4 signaling axis have a different or more prominent role in different breast cancer subtypes?

2. The authors should supplement with other cell lines, for example using conditioned media on breast cancer cell lines representing different subtypes such as MCF-7 (ER-positive, PR-positive) or SKBR3 (HER2-positive) to test whether the growth-promoting effect depends on the breast cancer subtype.

3. The study used the pharmacological inhibitor AS1517499 to inactivate STAT6. Small molecule inhibitors can sometimes have off-target effects. To increase the robustness of the results, did the authors consider confirming the results using genetic methods, such as using shRNA or siRNA to knockdown STAT6?

4. The signaling pathway is presented in a linear direction: STAT6 → JMJD3 → IRF4. However, do the authors think it is possible that IRF4 or one of its target genes could negatively regulate the activity of STAT6 or JMJD3?

5. The study demonstrated that JMJD3 knockdown reduced M2 markers. However, the spectrum of macrophage differentiation is considered to be a continuum rather than a discrete M1/M2 state. The question is: does JMJD3 inhibition simply block the M2 differentiation pathway, or does it actively push macrophages towards M1 polarization?

6. The authors could examine M1-specific markers (e.g., iNOS, IL-12) in JMJD3 knockdown cells, as this would provide further insight into the role of JMJD3 as a “switch” that directs polarization.

7. JMJD3 is a histone demethylase that can act on many different genes, not just IRF4. The study focused on the JMJD3/IRF4 axis. So do the authors think that JMJD3 can regulate other genes involved in M2 macrophage function in parallel with regulating IRF4?

8. The authors suggest that STAT6 inhibition reduces JMJD3 expression, suggesting that STAT6 regulates transcription of the JMJD3 gene. However, this is only an indirect relationship. Experiments are needed to demonstrate whether STAT6 acts on the promoter region of the JMJD3 gene to activate transcription (Chromatin Immunoprecipitation - ChIP).

9. The study only used conditioned medium to demonstrate that M2 macrophages affect cancer cells through the soluble factors they secrete. However, direct physical interactions between the two cell types may also play an important role. Direct co-culture of macrophages with breast cancer cells could be used to assess whether the observed effects are enhanced.

10. The authors have not specified which factors secreted by M2 are primarily responsible for promoting cancer cell growth.

**Do you want your identity to be public for this peer review?** For information about this choice, including consent withdrawal, please see our For information about this choice, including consent withdrawal, please see our Privacy Policy .

Reviewer #1: **Yes:** Marco CraveiroMarco Craveiro

Reviewer #2: No

While revising your submission, please upload your figure files to the Preflight Analysis and Conversion Engine (PACE) digital diagnostic tool, https://pacev2.apexcovantage.com/ . PACE helps ensure that figures meet PLOS requirements. To use PACE, you must first register as a user. Registration is free. Then, login and navigate to the UPLOAD tab, where you will find detailed instructions on how to use the tool. If you encounter any issues or have any questions when using PACE, please email PLOS at . PACE helps ensure that figures meet PLOS requirements. To use PACE, you must first register as a user. Registration is free. Then, login and navigate to the UPLOAD tab, where you will find detailed instructions on how to use the tool. If you encounter any issues or have any questions when using PACE, please email PLOS at figures@plos.org . Please note that Supporting Information files do not need this step.. Please note that Supporting Information files do not need this step.

---

## [Author Response · Author response to Decision Letter 1]

24 Oct 2025

Dear editors and reviewers:

We would like to thank you for your thorough review and valuable comments of our manuscript, “JMJD3 regulates the M2-like macrophage polarization and promotes the growth of breast cancer cells via the STAT6/IRF4 axis”. We are genuinely grateful for the opportunity to revise our work based on your insightful comments and suggestions.

We have carefully revised the manuscript according to the reviewers’ comments and ensured that it now fully complies with the PLOS ONE style and formatting requirements.

In this submission, we include a tracked version of the revised manuscript showing all changes, a clean version of the manuscript without tracked changes, and a point-by-point “Response to Reviewers” document.

In accordance with the editorial request, we have removed all funding-related information from the manuscript. The funding statement is as follows: “This study was supported by the Natural Science Foundation of Zhejiang Province (Grant No. LQ24H200001), the Medical Science and Technology Project of the Zhejiang Provincial Health Commission (Grant Nos. 2023KY353, 2024KY462), and the Shaoxing City Health Science and Technology Plan (Grant No. 2022SY020). There was no additional external funding received for this study. The funders had no role in study design, data collection and analysis, decision to publish, or preparation of the manuscript.”

And we have revised the data availability statement as follows: “All relevant data are within the manuscript and its Supporting Information files.”

Thank you once again for your constructive feedback and for helping us improve the quality of our work. We look forward to your further evaluation of the revised manuscript.

Sincerely,

Lihong Zhang,

Shaoxing People’s Hospital, 568 Zhongxing North Road, Yuechen District, Shaoxing, Zhejiang, 312000, China.

E-mail: whb0575@163.com

Reviewer #1

The research article “JMJD3 regulates the M2 macrophage polarization and promotes the growth of breast cancer cells via STAT6/IRF4 axis” by Juan Lyu et al investigates the role of JMJD3 in promoting M2 macrophage polarization through the STAT6/IRF4 signaling axis and draws a correlation between breast cancer progression and M2 TAMs.

Using data from over 1,000 breast cancer patients from the Cancer Genome Atlas (TCGA), the authors found that high levels of M2 macrophages correlate with advanced clinical features (e.g., age >60, ER+/PR+ status, late-stage tumors) and poorer overall survival.

In vitro, the authors induced M2 macrophage-like polarization by exposing THP-1 monocytes to PMA, IL-4, and IL-13. These M2 macrophage-like cells showed increased expression of JMJD3, IRF4, and phosphorylated STAT6. Knockdown of JMJD3 or inhibition of STAT6 reduced the expression of IRF4 and M2 markers while overexpression of JMJD3 enhanced M2 polarization and further supported cancer cell growth.

Mechanistically, the study demonstrates that JMJD3 promotes M2 polarization through the STAT6/IRF4 pathway. STAT6 activation triggered by IL-4/IL-13 increases JMJD3 expression, which in turn enhances IRF4 transcription. Disruption of this axis using a STAT6 inhibitor AS1517499 or JMJD3 knockdown resulted in impaired M2 polarization. In turn, MDA-MB cancer cells exposed to conditioned media of polarized THP-1 cells whose STAT6/JMJD3/IRF4 pathway had been disrupted showed decreased proliferation and increased apoptosis.

MINOR REVISIONS

1．THP-1 is monocytic cell line that can be differentiated into macrophage-like cells. The authors should refrain from classifying differentiated cells as macrophages and instead classify them as macrophage-like. Point in case, macrophages do not typically produce IL-2. Therefore, the inclusion of IL-2 measurement in figure 2D and referring to IL-2 as an M2 marker is incorrect and should be revised. Typically, M2 macrophages are characterized by IL-10, IL-4, and IL-13 secretion.

Response:

We sincerely thank the reviewer for this important and constructive comment. We agree that THP-1 differentiated cells should be more accurately described as macrophage-like cells rather than macrophages, and we have revised the manuscript to refer to M1 and M2 macrophages as M1-like and M2-like macrophages, respectively, throughout the text.

Regarding IL-2, we have removed IL-2 measurement from Figure 2D and Figure 4C and revised the related text.

2．Line 199/200: please provide a reference for the statement.

Response:

We thank the reviewer for pointing this out. We have now added 4 appropriate references to support the statement at lines 199/200. The revised sentence reads as follows: “Previous studies have indicated that the polarization of M2-like macrophages is regulated by the JMJD3 and STAT6/IRF4 signaling pathways [20, 22-24]” (line 207/208 in revised clean manuscript).

3. How do the authors account for the practically neglectable differences in apoptosis of MDA-MB-231 cells when subject conditioned medium from polarized THP-1/macrophages overexpressing or lacking JMJD3?

Response:

We thank the reviewer for this valuable comment. Although the differences in apoptosis of MDA-MB-231 cells treated with conditioned medium from JMJD3-overexpressing or JMJD3-knockdown macrophage-like cells were relatively small, they were still statistically significant. This suggests that JMJD3 may exert a modest but measurable regulatory effect on the apoptosis of tumor cells through macrophage-derived factors.

The limited magnitude of the observed effect may be attributed to several factors. First, MDA-MB-231 cells represent a highly aggressive and apoptosis-resistant TNBC subtype, which could mask subtle biological effects even when significant signaling changes occur. Second, the conditioned medium model only reflects soluble factor-mediated interactions and lacks the three-dimensional architecture and complex cellular crosstalk of the tumor microenvironment, where JMJD3-mediated macrophage–tumor interactions may exert more pronounced effects in vivo.

To address this further, we plan to use cytokine arrays or Olink proteomics to identify the specific soluble mediators involved. Additionally, co-culture systems that allow both soluble and contact-dependent interactions will help better mimic the in vivo TME and clarify whether JMJD3-mediated macrophage–tumor cell communication differentially regulates apoptosis in other breast cancer subtypes

4. The underlying concept of the paper is the role of JMJD3 in catalyzing the demethylation of H3K27me3 and promoting gene transcription. However, at no point in their model of silencing or overexpressing JMJD3 the authors actually show the effects on H3K27me3 demethylation. The authors should consider doing so as it would further validate their observations and strengthen the paper.

Response:

We thank the reviewer for this insightful comment. In response, we have performed experiments to examine the effects of JMJD3 modulation on H3K27me3 demethylation. As shown in the revised manuscript (S1 Fig), silencing JMJD3 led to an increase in H3K27me3 levels, whereas JMJD3 overexpression resulted in a marked decrease, confirming its demethylase activity. These findings further support our proposed mechanism and strengthen the overall conclusions of the study.

5．In the proposed model (Figure 6) the authors include CD163. However, at no point in the paper CD163 is mentioned. As such it should be removed from the model. Furthermore, the inclusion of “Cytokine” is not helpful when the authors know the cytokines in question: IL4 and IL13. Please substitute cytokine by IL4 and IL13 and make sure to use different colors for the dots, so that it is easily understood that there are two cytokines involved. I also find that the blue arrows that indicate what processes are occurring are not very intuitive. The authors might want to consider improving that.

Response:

We thank the reviewer highlighting this issue. In the revised model (Fig 6), we have replaced CD163 with CD206, which is a more representative marker of M2-like macrophages and is consistent with the rest of our manuscript. Additionally, we have changed the M1 and M2 macrophages to M1-like and M2-like macrophages.

Furthermore, as suggested, we have substituted the generic “Cytokine” label with the specific cytokines IL-4 and IL-13, and used different colors to distinguish them. We have also improved the arrow design to make the indicated processes more intuitive. The updated Figure 6 and its legend have been revised accordingly.

6．Finally, the authors need to provide data points for figures 2B, 2F, 3H, 3J, 3L, 4C, 4G, 5A, 5B, and 5C. Data points should also be more visible in all the figures. Additionally, the authors need to provide information on how many times the essays were done, if we are looking at biological repeats or technical repeats.

Response:

We thank the reviewer for this important comment. In response, we have made the following revisions: 1) We have added individual data points to all of the figures, as requested. We have also increased the visibility of all data points across the figures to ensure clarity. 2) Repetition details: We have clarified in the Materials and Methods section that all experiments were performed at least three independent biological replicates. This information has been added to the revised manuscript for transparency and reproducibility.

We believe these changes have improved the rigor and presentation of our study, and we are grateful for the reviewer’s careful attention to these details.

Reviewer #2:

1. TCGA analysis showed higher M2 scores in ER-positive and PR-positive patients. However, the functional assay was performed on the MDA-MB-231 cell line (ER-negative, PR-negative). Would the authors could explain about this? Does the STAT6/JMJD3/IRF4 signaling axis have a different or more prominent role in different breast cancer subtypes?

Response:

We thank the reviewer for this insightful comment. We acknowledge that TCGA analysis indicated higher M2 scores in ER-positive and PR-positive breast cancer patients, while our initial functional assays were performed using MDA-MB-231 cells (ER-negative, PR-negative). To address this concern and assess whether the STAT6/JMJD3/IRF4 axis functions similarly in other breast cancer subtypes, we have now performed additional experiments using MCF-7 cells. These new data have been incorporated into the revised manuscript (S2 Fig) and the corresponding Results section.

Consistent with our findings in MDA-MB-231 cells, M2-like macrophages promoted proliferation and inhibited apoptosis in MCF-7 cells, and manipulation of JMJD3 similarly modulated these effects. These results indicate that the STAT6/JMJD3/IRF4 signaling axis plays a conserved role in promoting M2 macrophage-mediated tumor-supportive functions across different breast cancer subtypes.

2. The authors should supplement with other cell lines, for example using conditioned media on breast cancer cell lines representing different subtypes such as MCF-7 (ER-positive, PR-positive) or SKBR3 (HER2-positive) to test whether the growth-promoting effect depends on the breast cancer subtype.

Response:

We thank the reviewer for this constructive suggestion. To address this, we have performed additional experiments using the ER-positive MCF-7 cell line. Conditioned medium from M2-like macrophages similarly promoted proliferation and inhibited apoptosis in MCF-7 cells, consistent with our observations in MDA-MB-231 cells. These results indicate that the growth-promoting effects of M2-like macrophages via the STAT6/JMJD3/IRF4 axis are conserved across different breast cancer subtypes. The new MCF-7 data have been included in the revised manuscript (S2 Fig) and corresponding Results section.

3. The study used the pharmacological inhibitor AS1517499 to inactivate STAT6. Small molecule inhibitors can sometimes have off-target effects. To increase the robustness of the results, did the authors consider confirming the results using genetic methods, such as using shRNA or siRNA to knockdown STAT6?

Response:

We sincerely thank the reviewer for this insightful and constructive comment. We fully agree that shRNA- or siRNA-mediated STAT6 knockdown, would provide complementary evidence and further strengthen the robustness of our conclusions.

At the same time, we chose the pharmacological inhibitor AS1517499 because the primary function of STAT6 in macrophage M2 polarization is mediated by its phosphorylated, active form rather than total protein levels. AS1517499 specifically blocks IL-4-induced STAT6 phosphorylation without affecting total STAT6 protein [1], allowing us to directly assess its functional activation. In contrast, shRNA or siRNA reduces total STAT6 expression, impacting both phosphorylated and unphosphorylated forms, which could confound interpretation of the phosphorylation-dependent function.

AS1517499 also provides rapid and reversible inhibition at specific time points after stimulation, which is particularly suitable for studying dynamic signaling and immediate functional responses, whereas genetic knockdown requires longer-term modulation and may trigger compensatory mechanisms. Using AS1517499, we were able to directly correlate inhibition of STAT6 phosphorylation with downstream effects on JMJD3, IRF4, and M2 marker expression, establishing a clear mechanistic link between STAT6 activation and macrophage polarization.

We acknowledge that genetic knockdown would add further support and plan to employ shRNA/siRNA or targeted mutations (e.g., STAT6 Y641F) to validate the function of STAT6 in macrophage polarization in future studies.

4. The signaling pathway is presented in a linear direction: STAT6 → JMJD3 → IRF4. However, do the authors think it is possible that IRF4 or one of its target genes could negatively regulate the activity of STAT6 or JMJD3?

Response:

We sincerely thank the reviewer for this insightful comment. We fully agree that IRF4 or its downstream targets could potentially exert negative feedback on STAT6 or JMJD3. Exploring such feedback loops could further deepen our understanding of the regulatory network governing M2-like macrophage polarization.

In our current study, we focused on the positive activation cascade from STAT6 to JMJD3 to IRF4 in THP-1-derived M2-like macrophages during IL-4/IL-13-induced polarization. Nevertheless, feedback loops are common mechanisms by which cells fine-tune signaling and prevent overactivation. Future experiments, such as manipulating IRF4 expression to assess its effect on STAT6/JMJD3, performing ChIP or ChIP-seq analyses to further reveal direct binding of IRF4 to the regulatory regions of STAT6 or JMJD3, and conducting functional assays to determine the impact on M2 polarization and tumor-promoting activity, will be essential to validate our proposed mechanism.

We thank the reviewer again for highlighting this important aspect.

5. The study demonstrated that JMJD3 knockdown reduced M2 markers. However, the spectrum of macrophage differentiation is considered to be a continuum rather than a discrete M1/M2 state. The question is: does JMJD3 inhibition simply block the M2 differentiation pathway, or does it actively push macrophages towards M1 polarization?

Response:

We thank the reviewer for this insightful comment. We fully agree that macrophage polarization represents a continuum rather than a strict M1/M2 dichotomy. In our study, JMJD3 knockdown not only reduced the expression of M2-associated markers (CD206, ARG1, and IL-10) but also increased the levels of M1-related markers, TNF. These results suggest that JMJD3 inhibition may not only suppress M2 polarization but also promote a shift toward an M1-like phenotype. However, we acknowledge that additional markers and functional assays are needed to further validate this observation and to comprehensively define the polarization state.

In future studies, we plan to examine a broader panel of M1-specific genes (e.g., iNOS, IL-12) and perform functional assays such as cytokine secretion and phagocytosis, which will provide a more comprehensive view of how JMJD3 regulates the continuum of macrophage polarization.

6. The authors could examine M1-specific markers (e.g., iNOS, IL-12) in JMJD3 knockdown cells, as this would provide further insight into the r

---

## [Decision Letter · Decision Letter 1]

6 Jan 2026

JMJD3 regulates the M2-like macrophage polarization and promotes the growth of breast cancer cells via STAT6/IRF4 axis

PONE-D-24-45718R1

Dear Dr. Zhang,

We’re pleased to inform you that your manuscript has been judged scientifically suitable for publication and will be formally accepted for publication once it meets all outstanding technical requirements.

Kind regards,

Li Yang, M.D.

Academic Editor

PLOS One

Additional Editor Comments (optional):

Thanks for the authors' efforts to comprehensively improve your manuscript according to editor's and reviewers' comments. I am pleased to inform you that your paper can be accepted for publication now.

Reviewers' comments:

Reviewer's Responses to Questions

**Comments to the Author**

Reviewer #2: All comments have been addressed

Reviewer #3: All comments have been addressed

2. Is the manuscript technically sound, and do the data support the conclusions?

Reviewer #2: Yes

Reviewer #3: Yes

3. Has the statistical analysis been performed appropriately and rigorously?

Reviewer #2: Yes

Reviewer #3: Yes

4. Have the authors made all data underlying the findings in their manuscript fully available?

Reviewer #2: Yes

Reviewer #3: Yes

5. Is the manuscript presented in an intelligible fashion and written in standard English?

Reviewer #2: Yes

Reviewer #3: Yes

Reviewer #2: The authors have been very proactive and professional in addressing concerns and suggestions raised in the previous review. The revised manuscript and the authors' detailed "Reviewer Response" document have addressed nearly all of the Reviewers' comments.

The authors added new experiments that confirm the demethylase activity of JMJD3 by demonstrating its effect on H3K27me3 levels, which strongly validates the proposed mechanism.

The addition of functional data using the MCF-7 (ER-positive) breast cancer cell line resolves previous discrepancies with the TCGA findings and reinforces that the STAT6/JMJD3/IRF4 axis plays a tumor-preserving, tumor-supporting role across different breast cancer subtypes.

The authors have also revised the model (Figure 6) and ensured that all figures now include individual data points and clarified that all experiments were performed on at least three independent biological replicates, improving transparency and reproducibility.

The authors openly acknowledge the remaining limitations (such as the lack of direct ChIP evidence of STAT6-JMJD3 binding and the need for in vivo validation) and are committed to exploring these questions through future experiments.

I have no further comments.

Reviewer #3: You may consider a systems biology approach or choose to use computational methods in future. I HiC-seq would give a genome-wide assessment of chromatin modification which would be helpful in your case.

**Do you want your identity to be public for this peer review?** For information about this choice, including consent withdrawal, please see our For information about this choice, including consent withdrawal, please see our Privacy Policy .

Reviewer #2: No

Reviewer #3: No

---

## [Editor Report · Acceptance letter]

PONE-D-24-45718R1

PLOS One

Dear Dr. Zhang,

I'm pleased to inform you that your manuscript has been deemed suitable for publication in PLOS One. Congratulations! Your manuscript is now being handed over to our production team.

Kind regards,

on behalf of

Dr. Li Yang

Academic Editor

PLOS One